# Corrosion Damage to Joints of Lattice Towers Designed from Weathering Steels

**DOI:** 10.3390/ma15093397

**Published:** 2022-05-09

**Authors:** Vít Křivý, Zdeněk Vašek, Miroslav Vacek, Lucie Mynarzová

**Affiliations:** 1Department of Building Structures, Faculty of Civil Engineering, VSB—Technical University of Ostrava, 708 00 Ostrava, Czech Republic; vit.krivy@vsb.cz (V.K.); lucie.mynarzova@vsb.cz (L.M.); 2Liberty Ostrava a.s., 719 00 Ostrava, Czech Republic; zdenek.vasek@libertysteelgroup.com

**Keywords:** steel structures, crevice corrosion, lattice towers, bolted lap joints, weathering steel, experimental tests, numerical modelling

## Abstract

The article dealt with the load-bearing capacity and durability of power line lattice towers designed from weathering steel. Attention was paid in particular to the bolted lap joints. The article evaluates the static and corrosion performance of bolted lap joints in long-term operating towers, and also presents and evaluates design measures that can be applied in the design of new lattice towers, or in the reconstruction of already operating structures. Power line lattice towers are the most extensive realization of weathering steel in the Czech Republic. On the basis of the inspections carried out to evaluate the working life of the transmission towers in operation, it can be stated that a sufficiently protective layer of corrosion products generally developed on the bearing elements of the transmission towers. However, the development of crevice corrosion at the bolted joints of the leg members is a significant problem. In this paper, the corrosion damage of bolted joints was evaluated considering two basic aspects: (1) the influence of crevice corrosion on the bearing capacity of the bolted joint was evaluated, using experimental testing and based on analytical and numerical calculations; (2) appropriate design measures applicable to the rehabilitation of developed crevice corrosion of in-service structures, or the elimination of crevice corrosion in newly designed lattice towers, was evaluated. Calculation analyses and destructive tests of bolted joints show that the development of corrosion products in the crevice does not have a significant effect on the bearing capacity of the joint, provided that there is no significant corrosion weakening of the structural elements, and bolts of class 8.8 or 10.9 are used. The results of the long-term experimental programme, and the experience from the rehabilitations carried out, show that, thanks to appropriate structural measures, specified in detail in the paper, the long-term reliable behaviour of the lattice towers structures is ensured.

## 1. Introduction

Weathering steels have been used in the Czech Republic for steel structures since the mid-1970s. Bridges and electric power transmission systems are among the typical objects constructed from it [1]. Under appropriate atmospheric and design conditions, weathering steels form a layer of protective oxides on their surface that significantly slows the corrosion rate [2,3]. Weathering steels are used for structures with a design working life of up to 100 years, without additional corrosion protection measures [4]. These steels are mainly known by the trade name Corten [5], of which Atmofix steels are a similar variant [6]. For structures not affected by chloride deposition in coastal zones, two grades of weathering steels are practically applied—S355J2WP (Corten A) and S355J2W (Corten B).

With the correct application of weathering steels, it is possible to apply and utilise a number of significant advantages in the process of realization and long-term operation of structures made of these materials, in comparison with structures made of other grades of structural steels that need protection against corrosion in the long-term, using traditional corrosion protection systems. These potential advantages are summarised and expressed in the following basic terms:When manufacturing and assembling structures made of weathering steel, it is usually possible to reduce the scope of some of the main production operations and, as a result, partially reduce the price of the structure realization. In particular, costs can be saved on the corrosion protection coating systems. On the other hand, the final price is partly increased by the increased costs of the initial rolled material and welding consumables, or possibly by the costs associated with the necessary corrosion allowances. The resulting total cost for a newly manufactured and assembled weathering steel structure is typically 2 to 10% less than a similar structure designed from other structural steels protected by a corrosion protection coating system [7].The possible savings for the implementation of corrosion protection coating treatments, according to the previous paragraph, are directly followed by the possibility of achieving significant savings in the time required for the realization of the construction. This advantage is significant and important, especially for investors and contractors of large structures, such as motorway bridges [8].The essential, and most important, advantage of using weathering steels is a significant reduction in the amount of labour, time, and cost required to ensure the inspection and necessary maintenance of long-term structures [9]. For weathering steel structures, the range and cost of these important activities is relatively small compared to the range and cost of a complete corrosion protection restoration of a comparable structure made of other structural steels [10]. In the case of weathering steel structures, inspection and maintenance are mainly the observation of the conditions required for the protective function of the patina, in particular the necessary surface cleanliness of the material in all elements of the structure and, if needed, also carrying out the necessary local repairs, and providing corrosion protection in corrosion-damaged details of the structure where a sufficiently protective and stable patina has not formed [11].Removing the residues of old corrosion protection, and applying a new coating system when revitalising old structures protected by coating systems or metallization, are operations that can be harmful to the health of workers and the surrounding environment. Ensuring the environmentally sound execution of a repair or resurfacing is technically challenging, and increases the cost of execution. The elimination of these operations due to the use of weathering steels, therefore, represents an undeniable advantage for health and environmental protection [12].The natural appearance and dark brown or dark purple colour of the stable patina may be advantageous in relation to the colour of the surrounding environment or vegetation [13]. The ability to naturally modify partial irregularities or damage created in the texture or colour of the patina is also advantageous.

Steel lattice towers are typically used as bearing structures in electric power transmission systems. These structures represent the most extensive realization of weathering steel in the Czech Republic. Between 1974 and 1992, approximately 4000 transmission towers, and 130 substations, of 110 kV, 220 kV, and 400 kV were built. Power line lattice towers are manufactured in various shapes, always in the form of four-sided lattice structures, and braced in the individual faces by bracing members made of simple angles. The leg members [14] of the angles are lapped along the height by bolted lap joints, and the bracing angles are connected to the leg members by a single bolt [15].

The corrosion behaviour of the weathering steels of transmission towers has been systematically evaluated since 1975 [16,17,18]. In general, the protective corrosion products develop favourably on the bearing elements of the transmission towers, even on surfaces not exposed to direct weathering.

However, for the use of weathering steels, the same details were originally applied as for zinc-galvanized or paint-protected transmission towers. No specific requirements were given for the operational control and maintenance of the transmission towers. Therefore, inadequately designed details lead to the unfavourable development of corrosion products [19,20]. These are mainly the following details of the lattice towers bearing structure:Anchoring the structure to concrete foundations;Bolted joints of leg members.

Corrosion weakening of the bearing elements in the area of transition to the concrete foundation is a typical failure of all structures, not only of weathering steel transmission towers. There are known cases where the critical weakening of the structure in the area of transition to the concrete foundation was one of the main causes of the collapse of power line towers.

In the case of the bolted joint of the leg members, crevice corrosion [21,22] occurs in the crevice, between the splices and the angles to be connected [23], as illustrated in Figure 1. The time of surface wetting is prolonged in the crevice, impurities accumulate, and aeration and concentration differences are created, all of which favour the development of the corrosion process. The conditions necessary for the formation of a protective patina do not occur in the crevice. The accumulated corrosion products deform the splices in the connection.

Concern about the possible effects of adverse corrosion development on the bolted joints of leg members led transmission system operators in the Czech Republic to abandon the concept of using weathering steels in the construction of new power lines. In order to operate existing transmission powers responsibly, it is important to have verified technical data on the effect of crevice corrosion on the bearing capacity of the joints. In this paper, the results of loading tests of the leg members joints of two 110 kV power line transmission towers, taken from collapsed transmission towers, are presented. The main objective of the experimental programme, and subsequent computational analyses, was to verify the effect of crevice corrosion on the mechanical resistance of the transmission tower. The calculation analyses of bolted joints are performed using analytical equations, according to current standards [24,25]. For detailed numerical analyses of bolted joints using the finite element method, the recommendations given in technical publications [26,27,28] can be used.

High-voltage power lines with weathering steel transmission towers are still in operation. The transmission system operators carried out extensive rehabilitation work on most of the transmission towers, including the rehabilitation of the bolted joints in the lattice structure of the transmission towers. One of the objectives of the authors of the article was to provide concentrated information on the real technical condition of the operating transmission towers. The specific findings and recommendations are based primarily on the authors’ long-term experience with the design, realization, and reliability assessment of these structures. The article deliberately provided mainly practically applicable information and recommendations for repair and maintenance.

In order to take advantage of the economic and environmental benefits of using weathering steels in the construction of lattice towers in the future, it is necessary to obtain demonstrable data confirming the functionality of the recommended design solutions for the bolted joints of leg members. For this purpose, a programme of experimental lattice towers was prepared, on which various bolted joints designs were tested in real conditions. After 9 years of the experimental programme, sufficiently reliable data are now available to confirm the suitability of the various design measures.

The issue of the corrosion behaviour of structures designed from weathering steels is very extensive, and this area received attention from researchers all over the world. Most of the scientific literature is devoted, in detail, to the study of the evolution of corrosion layers under different environmental conditions [29,30], or using accelerated tests [31,32]. The development of appropriate prediction models is important for the design of building structures [33]. This article, however, aimed at a somewhat different goal. The focus was mainly on the evaluation of the real influence of the development of corrosion products on the load-bearing capacity and durability of lattice truss towers with bolted joints, and on the design of appropriate structural measures. The presented results can be used by the scientific community, but also by experts from the construction industry, who are responsible for the design or operation of steel structures of power line transmission systems.

## 2. Materials and Methods

### 2.1. Loading Tests of Bolted Joints

The material for the test specimens was taken from the leg members of the collapsed power line transmission towers of 2 110 kV power lines *Neznášov—Týniště nad Orlicí* (overhead lines V1195 and V1196 in the Czech Republic). The transmission towers were made in the 1980s, using the weathering steel Atmofix made by Vítkovice Ironworks, Czech Republic. Eight test specimens were subsequently prepared from the collected leg members.

For the destructive tensile test of the joints, two pieces of the A-type joint, and six pieces of the K-type joint, were prepared. In all cases, L90/6 equal-leg angles were connected using P8 splices. The splices were connected to the profiles with 2 × 6 M20 bolts, of the class 5.6. The geometry of the test specimens is presented in Figure 2, Figure 3 and Figure 4.

The chemical composition of the steel was determined using an optical emission spectrometer. Chemical analysis was carried out on two selected angles (test specimen A1 and K1), and one splice (test specimen K1). The chemical analysis confirms that the specimens taken from the collapsed lattice towers correspond to standard weathering steels, in accordance with the requirements of the standards [34,35]. The results are presented in Table 1. The mechanical properties of the steel were tested in the accredited laboratory of the Brno University of Technology [36]. The tensile testing, according to EN ISO 6891-1 [37], was carried out on eight flat test pieces. For the tensile testing, five pieces of leg members and three pieces of splices were made. The results are given in Table 2. The results of the chemical analysis and mechanical properties are compared with the original Czech national standard CSN 41 5217 [34], and the currently valid European standard EN 10025-5 [35]. The material meets the chemical and mechanical requirements for Atmofix steel (S355J2W).

During the tensile testing of the bolted joints, the relation between the tensile force *F* (kN) and the total deformation *u* (mm) of the test specimens was registered. At the same time, the gap opening (i.e., the change in the longitudinal distance between the ends of the connected angles) between the connected members was also measured. The uniform length of the test specimens between fixed ends is *L*_0_ = 1010 mm, see Figure 5. The tensile testing was carried out until the specimen failed. Verification of the bearing capacity of the joint was performed using analytical relationships, in accordance with the valid European standards EN 1993-1-1 [24] and EN 1993-1-8 [25]. Numerical analysis was also carried out on the selected joint using ANSYS software. The joint was modelled using 3D solid elements and assuming the application of deformation load. The material properties of the steel and bolts were introduced using the Ramberg–Osgood stress–strain curve [38].

The microstructure of corrosion products in the crevice was analysed using a scanning electron microscope SEM and a EDAX analyser. The results are presented in Table 3 and Table 4.

### 2.2. Assessment of the Technical Condition of the Transmission Towers in Operation

The transmission towers that are part of very high-voltage power lines in the Czech Republic (overhead lines *V434 Slavětice—Čebín, V437/V438 Slavětice—Dürnrohr*) were selected for evaluation. The overhead lines were built in the 1980s, and have been in operation for approximately 40 years. In July 2021, an inspection of the steel bearing structure was carried out at 5 representative transmission towers, with a focus on the evaluation of the development of corrosion products in the area of the bolted joints of leg members.

The transmission towers are located in an agricultural area (corrosivity category C2). The structural design of the transmission towers can be seen from the photographs presented in Figure 6.

The transmission towers were visually inspected to evaluate the development of corrosion products, and the effectiveness of the implemented rehabilitation measures. The thickness of the corrosion products was measured on typical elements of the transmission towers structure, using the magnetic induction method with the Positector 6000 instrument. A total of 30 measurements were made on each of the evaluated surfaces. The actual thicknesses of the structural elements were continuously measured using a Positector UTG ME ultrasonic thickness gauge.

### 2.3. Experimental Lattice Towers

Experimental verification of the bolted joints in steel lattice structures designed from weathering steel was carried out in the premises of steelmaker Liberty Ostrava Inc. For this purpose, three experimental towers were built in 2012, on which the long-term monitoring of the development of corrosion products in the field of bolted joints with different structural design was carried out, as illustrated in Figure 7. This was a long-term experiment, the aim of which was to find a variant of the bolted joint of the leg members that minimized the formation of crevice corrosion in the long term, and at the same time be simple, inexpensive, and not complicate the assembly and maintenance of the structure.

During the experiment, the effectiveness of individual structural measures against the formation of crevice corrosion was continuously monitored. The design of the towers differed only in the joints of the leg members. In particular, the influence of the following factors was examined:The effect of spacings between centres of fasteners and of end, and edge, distances from the centre of a fastener in the connection. Connections with the minimum allowable spacing and end/edge distances were expected to be less susceptible to crevice corrosion. Therefore, for tower 1, all splices were designed with smaller spacings and end/edge distances, compared to towers 2 and 3 (for tower 1, the minimum permissible values were chosen in accordance with EN 1993-1-8).The effect of the thickness of the splice. It was assumed that joints with thicker splices are less susceptible to crevice corrosion. On towers 1 and 2, all splices were designed with a nominal thickness of *t* = 10 mm, and on tower 3 the splices were designed with a nominal thickness of *t* = 5 mm.The effect of bolt preloading. Joints with preloaded bolts were expected to be less susceptible to crevice corrosion. Half of the bolted joints on each tower included preloaded bolts (uncoated M20 bolts, strength class 8.8, bolt shank coated with Teflon Vaseline M8062, tightening moment 400 Nm).The effect of treatment of the contact surface of the splices and leg members. It was assumed that connections with treated contact surfaces are less susceptible to crevice corrosion. Four types of contact surfaces were tested—contact surface of splices without treatment, contact surface of splices with paint (FEST-B S2141 paint), contact surface of splices with silicone sealant (neutral silicone OXIM; the method of application of silicone sealant is documented in Figure 8), and the contact surface of splices with Vaseline (Teflon Vaseline M8062; the method of application of Teflon Vaseline is illustrated in Figure 9). There are always two joints on each lattice tower with the appropriate contact surface treatment—one joint with non-preloaded bolts, the other joint with preloaded bolts.

All towers have a rectangular ground plan of 1200 × 1500 mm, and the height of the towers is 2000 mm. Each tower has two vertical faces, and two sloping faces inclined from the vertical plane at an angle of 10°. The leg members are made of equal-leg angles L120 × 10, and the diagonals are made of equal-leg angles L50 × 5. Steel grade S355J2W was used. The angles used were produced on the rolling mills HCC and SJV of Liberty Ostrava Inc. The chemical composition of the leg members is listed in Table 5. The values of carbon equivalent, *CEV* = 0.42%, and atmospheric corrosion index, *I* = 6.3%, were calculated from the chemical composition [39,40].

The mechanical properties of the hot-rolled products are summarized in Table 6. The yield strength *R*_eH_ exceeds the characteristic value of 355 MPa by 20%, and the tensile strength is in the required range of 470–630 MPa. The A5 elongation value is 35%, above the required limit of 22%. Although the melt was rolled in S355J2W grade, the impact strength KV with 2 mm V-notch at 0 °C, −20 °C, and −50 °C was also determined. According to the results obtained, very good notch toughness of the tested steel is evident, even at extreme negative temperatures.

The diagonals of the towers were rolled on the rolling mill SJV of Liberty Ostrava Inc. The results of mechanical properties are listed in Table 7. The lower values of impact strength, compared to the L120 × 10 equal-leg angles, are due to the smaller non-standard tested thickness. However, the minimum value of 13.5 (J) is still met.

The splices were also rolled on the rolling mill SJV as P80 × 10 and P60 × 10 flat pieces, and L80 × 5 angle. The mechanical properties of the splices are provided in Table 8.

According to the analyses of chemical composition and mechanical properties, good compliance with the requirements of the EN 10025-5 standard for grade S355J2W can be stated. The experimental towers were built from material that meets the conditions of weathering steels of the required strength grade. All joints were designed with bolts. There are two bolted lap joints on each leg member. The towers are located in the premises of Liberty Ostrava Inc. (the area corresponds to the corrosivity category C3).

The surface of the angles and splices was left as rolled, including the mill scales. Only the coated splices were cleaned down to clean metal.

## 3. Results

### 3.1. Loading Tests of Bolted Joints

#### 3.1.1. Calculation of the Resistance of the Bolted Connection

To compare with the results of the loading tests, the following section calculates the resistance of the bolted connection, according to the European standards EN 1993-1-1 and EN 1993-1-8.

The calculation is carried out for a K-type joint. Specimens for the loading tests are taken from transmission lattice towers made of Atmofix steel, which correspond to the current structural steel grade S355J2WP (yield strength *f*_y_ = 355 MPa, ultimate strength *f*_u_ = 510–680 MPa). The leg members of the L90 × 6 equal-leg angles are connected. In the connection, double-sided P8 × 90 splices are designed. The M20 non-preloaded bolts, of strength class 5.6, are used in the connection. The hole diameter for the bolts is *d*_0_ = 22 mm. The bolt spacing and the end and edge distances are shown in Figure 3. The resistance of the following components of the connection is determined in sequence:Angle subjected to tension (calculation of *N*_pl,Rd_ and *N*_u,Rd_ for the angle);Contact splices subjected to tension (calculation of *N*_pl,Rd_ and *N*_u,Rd_ for the splices);Block tearing of a bolt group out of the angle (calculation of *V*_eff,2,Rd_);Bolts subjected to shear (calculation of *F*_v,Rd_ and *F*_b,Rd_).

The design plastic resistance of the gross cross-section of the angle is determined, according to EN 1993-1-1, cl. 6.2.3(2a):(1)$$N_{\mathrm{pl},\mathrm{Rd}}=\frac{A\;f_{y}}{\gamma_{M0}}=\frac{1050\cdot 355}{1.0}\cdot 10^{-3}=372.8\;\mathrm{kN}$$

The design ultimate resistance of the net angle cross-section at holes for fasteners is determined, according to EN 1993-1-1, cl. 6.2.3(2b):(2)$$N_{u,\mathrm{Rd}}=\frac{0.9\;A_{\mathrm{net}}\;f_{u}}{\gamma_{M2}}=\frac{0.9\cdot 816\cdot 510}{1.25}\cdot 10^{-3}=299.6\;\mathrm{kN}$$

The design plastic resistance of the gross cross-section of the splices is determined, according to EN 1993-1-1, cl. 6.2.3(2a):(3)$$N_{\mathrm{pl},\mathrm{Rd}}=\frac{A\;f_{y}}{\gamma_{M0}}=\frac{1440\cdot 355}{1.0}\cdot 10^{-3}=511.2\;\mathrm{kN}$$

The design ultimate resistance of the net splice cross-section at holes for fasteners is determined, according to EN 1993-1-1, cl. 6.2.3(2b):(4)$$N_{u,\mathrm{Rd}}=\frac{0.9\;A_{\mathrm{net}}\;f_{u}}{\gamma_{M2}}=\frac{0.9\cdot 1088\cdot 510}{1.25}\cdot 10^{-3}=399.5\;\mathrm{kN}$$

The design block shear tearing resistance for a bolt group out of the angle is determined, according to EN 1993-1-8, cl. 3.10.2:(5)$$V_{\mathrm{eff},2,\mathrm{Rd}}=2\left(\frac{0.5f_{u}A_{\mathrm{nt}}\;}{\gamma_{M2}}+\frac{\frac{f_{y}}{\sqrt{3}}A_{\mathrm{nv}}}{\gamma_{M0}}\right)=2\left(\frac{0.5\cdot 510\cdot 204\;}{1.25}+\frac{\frac{355}{\sqrt{3}}\cdot 870}{1.0}\right)\cdot 10^{-3}=439.9\;\mathrm{kN}$$

The design shear resistance (shear plane passes through the unthreaded portion of the bolt) is determined, according to EN 1993-1-8, cl. 3.4.1, 3.6.1, and 3.7:(6)$$F_{v,\mathrm{Rd}}=n\frac{\alpha_{v}f_{\mathrm{ub}}A\;}{\gamma_{M2}}=6\cdot \frac{0.6\cdot 500\cdot 314}{1.25}\cdot 10^{-3}=452.2\;\mathrm{kN}$$

The design bearing resistance is determined, according to EN 1993-1-8, cl. 3.4.1, 3.6.1, and 3.7:(7)$$F_{b,\mathrm{Rd}}=n\frac{k_{1}\alpha f_{u}dt\;}{\gamma_{M2}}=6\cdot \frac{2.5\cdot 0.606\cdot 510\cdot 20\cdot 6}{1.25}\cdot 10^{-3}=445.0\;\mathrm{kN}\;$$

On the basis of a comparison of the resistances corresponding to the possible failure modes of the bolted joint of the leg member, it can be stated that the design ultimate tension resistance of the net angle cross-section at holes for fasteners determines the resistance of the joint $N_{u,\mathrm{Rd}}=299.6\;\mathrm{kN}$.

#### 3.1.2. Analytical Calculation of the Tensile Force in Bolts Due to Pressure of Corrosion Products in the Crevice

Bulky corrosion products develop in an imperfectly sealed crevice of a bolted joint. The corrosion products formed push the connected parts away from each other, so that the crevice opens up. The separation of the connected members is prevented by the bolts, where tensile forces are generated. The experiments carried out show that at the bolt location the gap opening is zero, and that with increasing distance from the bolts, the gap opening of the connection gradually increases. A permanent deformation of the connected members occurs.

A conservative estimation of the tensile force acting on the bolt *F*_t,Ed_ can be determined on the basis of the assumption of plastic loading of the splice. For the calculation of the plastic bending resistance of the splice, the bolt spacing *L* = *p* = 80 mm, the splice width *b* = 90 mm, and the flange thickness *t* = 8 mm are considered. A simple beam model with uniform loading is assumed, and the tensile force in the bolt is equal to the reaction:(8)$$M_{\mathrm{pl},\mathrm{Rd}}=\frac{W_{\mathrm{pl}}f_{y}}{\gamma_{M0}}=M_{\mathrm{Ed}}=\frac{1}{8}q_{z}L^{2}$$
(9)$$F_{t,Ed}=q_{z}L=\frac{2\;bt^{2}f_{y}}{\gamma_{M0}L}=\frac{2\cdot 90\cdot 8^{2}\cdot 355}{1.0\cdot 80}\cdot 10^{-3}=51.1\;\mathrm{kN}$$

The bolts in the connection are assessed for combined shear and tension, according to EN 1993-1-8, cl. 3.6.1. The shear force into the bolt is determined from the resistance of the least bearing member of the connection, i.e., the design ultimate resistance of the net angle cross-section at holes for fasteners:(10)$$F_{t,\mathrm{Rd}}=n\frac{k_{2}f_{\mathrm{ub}}A_{s}}{\gamma_{M2}}=\frac{0.9\cdot 500\cdot 245}{1.25}\cdot 10^{-3}=88.2\;\mathrm{kN}$$
(11)$$\frac{F_{v,\mathrm{Ed}}}{F_{t,\mathrm{Rd}}}+\frac{F_{t,\mathrm{Ed}}}{1.4\;F_{t,\mathrm{Rd}}}=\frac{49.9}{75.4}+\frac{51.1}{1.4\cdot 88.2}=0.662+0.414=1.076>1.0$$

The above assessment concludes that the combined action of extreme load effects, due to the axial stresses in the leg member, and the maximum tensile effects in the bolts, caused by the development of bulky corrosion products in the crevice, leads to bolts failure. This finding is valid for bolts of lower strength classes (4.6, 5.6) that were previously used in the design of transmission towers. Nowadays, bolts with strengths classes of 8.8 or 10.9 are commonly used, which minimizes the possibility of bolt failure in the connection under evaluation. 

#### 3.1.3. Results of Loading Tests

The loading tests are carried out until the failure of the test specimens, which occurs either by rupture of the angle of leg member, or by shear of the bolts. Summary results for both ‘A’ and ‘K’ test specimens are shown in Table 9. The ultimate load at failure of the test specimen is in the interval 441 to 484 kN, the average value being 466.9 kN. The load at the beginning of the plastic deformation of the test specimen in all cases is close to 350 kN (visible reduction of the axial stiffness of the specimens, as shown Figure 10). Significant differences are observed in the deformation values of the individual test specimens. In the opinion of the authors of the paper, these differences are mainly due to the different values of slip in the individual joints, and probably also the different levels of friction in the joints. The failure of the test specimen occurs by rupture of net cross-sectional area at holes for fasteners, as illustrated Figure 11. In one case, there is a shear failure of the bolts.

#### 3.1.4. Results of Numerical Models

Numerical analysis is performed for the K-type joint in ANSYS software. Both cases of possible failure are considered, i.e., without the influence of the crevice corrosion, and then with the influence of the crevice corrosion on the joint taken into account. The models are loaded with deformation loads. In the lower part of the model, a support with zero displacements is placed on the L90 × 6 profile, and a linear displacement is applied to the upper cross-sectional area, in a direction parallel to the specimen under load. The magnitude of the displacement corresponds to the measured real magnitude (i.e., the average of the measured *u*_max_ displacement values given in Table 9; i.e., 22.1 mm). The model mesh has a size of 2 mm (the major part is created using tetrahedron and hexahedron type of mesh; the minor part is created using the triangular prism and pyramid type of the mesh). Linear finite elements are used in the numerical model. The number of nodes in the model is 293,652. For contact between bodies, the *ANSYS contact tool* is used, with friction (a coefficient of friction 0.15).

For the K-type joint modelled without the effect of joint corrosion, the maximum von Mises equivalent stress of 497.7 MPa is obtained by numerical analysis. Figure 12 and Figure 13 demonstrate the distribution of the von Mises equivalent stress, and the von Mises equivalent strain, and, therefore, the assumed location of the real failure is confirmed by destructive tests. The reaction force of the model is 474.0 kN, and the average value measured during destructive testing is 466.9 kN.

For a K-type joint modelled considering the effect of crevice corrosion, the von Mises equivalent stress of 499.9 MPa is obtained by numerical analysis. The consideration of the effect of crevice corrosion is based on a specific case of deformation of the splice of the K2 tested joint, where the measured maximum size of the crevice with corrosion products is equal to 10 mm. The model prescribes a 10 mm deformation of the vertical lines of the splices away from the L profile. The total elongation of the assembly is set, according to the K2 failure test, to 15 mm. The reaction force of the model is 486.7 kN, and the real measured force in the destructive test with bolt failure is 475 kN. Figure 14 and Figure 15 below show, on the left, the results corresponding to the application of deformations from the crevice corrosion, and on the right, the results after the application of the subsequent load by the prescribed displacement.

These results show that the bolts are significantly stressed from the beginning of the development of corrosion products in the crevice. With subsequent static tensile loading, the stresses in the bolts increase further. The numerical analysis is in good agreement with the results of the analytical calculation presented in Section 3.1.2. 

### 3.2. Assessment of the Technical Condition of the Transmission Towers in Operation

#### 3.2.1. Development of Corrosion Products on Leg Members and Diagonals

A protective adhesive compact layer of corrosion products, typical of directly wetted surfaces, develops on the surface of all the evaluated transmission towers. The average thickness of corrosion products is a suitable qualitative indicator of the favourable development of corrosion products on structures designed with weathering steels. The results of the thickness of corrosion products measurements (see Table 10) confirm the findings of the visual inspection. The thicknesses of the corrosion layers are similar on all the assessed transmission towers. The average corrosion thicknesses range from 136.3 μm to 197.7 μm, with a mean value of 161.0 μm. Thus, the criterion reported in the literature, [41,42] defining a maximum average thickness of corrosion products of 400 μm for sufficiently protective patinas, is reliably fulfilled. The value of the coefficient of variation ranges from 0.17 to 0.29, with a mean value of 0.22.

#### 3.2.2. The Bolted Joints of the Leg Members 

For all lattice towers situated at sites locality 1 to locality 5, as presented in Figure 6, the development of corrosion products in the crevice between the angle and the splices is identified. The bolted joints rehabilitation had not yet been performed on the transmission tower placed at locality 5. In the area of the bolted joint of the leg member, a typical development of corrosion products in the crevice between the splices and the leg member is observed, associated with the development of permanent plastic deformation of the splices, as illustrated in Figure 16. The thickness of corrosion products in the crevice reaches up to 10 mm.

For the other transmission towers located at sites locality 1 to locality 4, the rehabilitation of the bolted joints was already carried out. The rehabilitation of the transmission towers was undertaken approximately 10 years ago. The repair of the transmission towers is carried out by workers wearing full body harnesses, and who are authorised to work at heights. In these difficult working conditions, it is very complicated to comply flawlessly with all the requirements recommended for the rehabilitation of the bolted joints of the leg members. For the transmission towers evaluated, failures of the additionally applied coating system are more frequently identified, especially on surfaces that the primer was applied to without prior proper surface cleaning (refer to Figure 17). However, from the point of view of the long-term reliable functioning of the structure, these failures of the coating system do not represent a significant problem, as the original protective layer of corrosion products remain under the peeling coating.

The protection of the crevice by the sealant is still functional, and no significant failures are identified on the transmission towers evaluated (see Figure 18). Thus, the bolted joint of the leg members is protected from further development of crevice corrosion, even after about 10 years following the application of the sealant.

### 3.3. Experimental Lattice Towers

#### 3.3.1. Development of Corrosion Products on Leg Members and Diagonals

The development of corrosion products on leg members and diagonals is monitored at regular intervals. The increase in the average corrosion thicknesses is noted in Figure 19. It is expected that the average thickness of the corrosion products will continue to increase gradually in the coming years (for the long-standing transmission towers in operation, an average thickness of corrosion product of 161 μm is found, see Chapter 3.2.1). During the manufacture of the towers, the surface of the steel elements is deliberately left as rolled, including the mill scales. The thin surface mill scale has mostly fallen off after 9 years of exposure. The rolled thicker scales still remain on some surfaces, but their surface is gradually diminishing. 

Statically significant corrosion weakening is not yet observed; the differences in the thickness of the flanges at the time of installation of the lattice towers and after 9 years of exposure are minimal. As the experimental lattice tower project is designed to be long-term (assumed to be at least 25 years), detailed analyses of corrosion products sampled from the crevice of the de-installed bolted joints are not yet available.

#### 3.3.2. The Bolted Joints of the Leg Members

No phenomena related to the formation of crevice corrosion are observed in tower 1, which is designed with 10 mm thick splices, and with minimum bolt spacing and end/edge distances. No deterioration of the sealant is observed in the connections protected by sealant, as shown in Figure 20. For the other connection types, no mechanical damage to the splices is observed so far, although a lighter strip of corrosion products is visible on the top edge of the connection. No differences are observed between joints with preloaded and non-preloaded bolts.

For tower 2, which is designed with 10 mm thick splices, and with normal bolt spacings and end/edge distances, phenomena related to the formation of corrosion products in the crevice between the bolt and the leg member are already observable. For bolted joints without treatment, the initial development of corrosion products in the crevice are observed (the thickness of the corrosion products at the top edge of the splice is still very small: up to about 0.5 mm, as demonstrated in Figure 21). For the other connections, the development of crevice corrosion is not yet identified, although a lighter strip of corrosion products is visible at the top edge, especially for the joints with coatings. All joints protected with sealant are functional. No significant differences between preloaded and non-preloaded bolts are observed. There is a visual influence on the development of corrosion products around the joints protected with Vaseline (this is only a visual failure, with a slowing of the development of corrosion products).

In tower 3, which is designed with 5 mm thick splices, and with normal bolt spacings and end/edge distances, development of crevice corrosion occurs at some bolted joints, as illustrated in Figure 22. For untreated bolted joints, corrosion products with a thickness of up to 3 mm are found at the top edge of the splice (the thickness of corrosion products is slightly less for preloaded bolted joints, approx. 2 mm).

Corrosion products with a thickness of 1 mm are also found on joints (both non-preloaded and preloaded bolted joints) protected by coating system. The joints protected with sealant are functional, as indicated in Figure 23.

## 4. Discussion

The environmental and economic benefits of using weathering steels for power line constructions can only be realised if the necessary technical data on the long-term behaviour of the structural elements, and their connections, are available. Therefore, in this paper, attention was paid to the issue of long-term reliable functioning of bolted joints of leg members, in relation to the possible development of corrosion products in the crevice between angles and splices. This paper examined the bolted joints of leg members from two basic aspects:Verification of the static resistance of bolted joints where crevice corrosion has already developed;Design and long-term verification of structural measures that eliminate the occurrence or further development of crevice corrosion in the long term.

### 4.1. Verification of the Static Resistance of Bolted Joints

The tension resistance of the bolted joint of the leg members was verified experimentally, on undamaged specimens taken from the collapsed transmission towers (the cause of the collapse of the towers was corrosion damage in the area of anchoring the leg members to the foundation). Analytical and numerical calculations of the resistance of the selected connection were performed for comparison. The performed calculations are in good agreement with the results of the loading tests, where in most cases the failure of the test specimen occurred due to tensile damage of the net angle cross-section at holes for fasteners. In only one case was the shear of the bolts responsible for the failure of the specimen (in this case, the increased tensile stress on the bolts from the pressure of bulky corrosion products in the crevice may have contributed to the failure mode). For all tested specimens, the value of the elastic capacity of the joint $F_{el}≅350\;\mathrm{kN}$ is higher than the resistance calculated, according to the applicable standards $F_{T,Rd}=N_{u,\mathrm{Rd}}=299.6\;\mathrm{kN}$.

On the dismantled joints used for testing, it is observed that the thickness of the corrosion layer is greatest around the perimeter of the splice, decreasing towards the bolts. There is no significant corrosion weakening of the bolts. Thus, permanent deformation of the splices does not have a significant effect on the shear resistance of the bolts. This observation is consistent with the results of earlier studies carried out on bolted lap joints made of weathering steel [17,43].

The calculations presented in Section 3.1.2, as well as the numerical analyses presented in Section 3.1.4, indicate a possible negative effect related to the increase in tensile stresses on the bolts, caused by the development of corrosion products in the crevice. This process represents the direct effect of corrosion development in the crevice on the static load capacity of the structural member. When the extreme load effects from the axial stresses in the leg member are combined with the maximum tensile effects in the bolts, caused by the development of bulky corrosion products in the crevice, failure of the bolts may occur, provided that the bolts are designed using lower grades steels. This conclusion follows from the quantification of Equations (8)–(11).

The calculation analyses given in Section 3.1.1, Section 3.1.2 and Section 3.1.4 do not include possible corrosion weakening of the cross-sections of the individual components of the joint. It is assumed that possible corrosion weakening may have the following consequences for the resistance of the connection:The shear and bearing resistance of the bolts will be minimally affected, because the bolt shank is located inside the connection, and the corrosion weakening of the bolt head and nut does not have a significant effect on the way the leg members are stressed.The greatest corrosion damage to the splices is expected in the gross cross-section between the bolts. However, the splices satisfy the resistance criterions with a margin in this cross-section, because they are not weakened by the bolt hole. In the net cross-section at the bolt, the effect of corrosion is less than in the cross-section between the bolts, due to the bolt tightening the crevice. The results of residual thickness measurements in the joint are found in [17], and the least corrosion weakening was found in the circumference of bolt holes.The corrosion weakening of the leg members designed from angles is similar to that of the contact splices. As a result of the smaller cross-sectional area, and the smaller thickness of the leg member, this effect may be more pronounced than for splices, and should always be assessed according to the specific condition of the towers. Therefore, the angle of the leg member is likely to remain the weakest component of the joint when considering corrosion weakening.

The results of the experimental and computational analyses of the bolted joints of the leg members correspond to the experience with the operation of the transmission towers. The cause of accidents of lattice towers designed from weathering steels in the Czech Republic is usually the loss of stability of the compressed elements of the lattice structure subjected to increased bending stresses (e.g., from the effects of wind loading), or significant corrosion damage in the area of the anchorage of the transmission tower to the concrete foundation. Adverse microclimatic conditions can occur in the anchorage area: the structure is affected by surrounding vegetation; structural elements are often permanently covered with soil; and frequent degradation of the foundation structures also contributes to increased wetting. In the case of transmission towers designed from weathering steel, a continuous cycle of wetting and drying of the surface is not ensured in locally affected areas and, therefore, a protective layer of corrosion products does not develop on the structural elements.

This results in corrosion weakening the bearing elements in the anchorage area. Selected typical examples of corrosion damage in the anchorage area of transmission towers are provided in Figure 24.

### 4.2. Design Solutions for New Transmission Towers

The experimental towers located on the premises of Liberty Ostrava Inc. are planned as a long-term corrosion test, which focuses primarily on the verification of design measures to eliminate the development of corrosion products in the crevice at the bolted joints of the leg members. Based on the results obtained after 9 years of exposure, the functionality of the individual design measures can already be reasonably evaluated.

The basic design measure is the design of sufficiently rigid splices, together with the use of minimum recommended bolt spacing and end/edge distances. Both of the above principles are applied to tower 1, where corrosion products at the crevice in any of the bolted joints evaluated have not yet been observed to develop. The effectiveness of the proposed measure is based on a basic static assumption. By increasing the thickness of the splices, and reducing the spacing and end/edge distances, a component with higher bending stiffness is installed in the bolted joint, which is able to resist transverse pressures from corrosion products arising in the crevice.

The application of silicone sealant seems to be an appropriate measure. The principle of the method is to seal the edge between the splice and the angle against the direct effects of the atmosphere. The application of silicone sealant on the splice takes a maximum of 30 s, and the sealant is cheap. After placing the splice into the structure, and the subsequent tightening of the bolts, the sealant is pushed beyond the edges of the splice for both non-preloaded and preloaded joints, and the sealant is easily smoothed around the perimeter of the splice. After 9 years of exposure, all the sealant-coated joints are functional, and the sealant intact, with no signs of cracking or other degradation. The suitability of this design measure was also verified for operational transmission towers, where crevice sealing was one of the partial steps in the rehabilitation of bolted joints affected by the development of crevice corrosion (see Section 3.2.2). The use of sealants for the renovation of bolted joints is also recommended in the guidelines for the use of steels with increased resistance to atmospheric corrosion [42].

Based on the results obtained after 9 years of exposure of the towers, it is not possible to clearly evaluate the beneficial effect of bolt preloading. However, the results obtained from tower 3 suggest that bolt preloading in the connection reduces the risk of the development of corrosion products in the crevice. The beneficial effect of preloading is documented by mathematical numerical models [26,27]. The real effect of bolt preloading in weathering steel lattice towers is also planned to be monitored in detail in the remaining years of the experimental measurement.

On the other hand, measures based on coating the contact surfaces of the splices appear to be less effective. Although the development of bulky corrosion products in the crevice has not been observed in bolted joints with Teflon Vaseline-protected splices after 9 years of exposure, the adverse visual effect on the development of corrosion products around the connection significantly limits the applicability of this measure.

### 4.3. Recommended Method of Rehabilitation of Bolt Joints of the Leg Members

While a sufficiently protective layer of corrosion products usually develops on the common elements of the steel bearing structure of weathering steel lattice towers, locally unfavourable behaviour related to the development of crevice corrosion occurs in the bolted joints areas. Loading tests carried out on specimens taken from the collapsed transmission towers show that the static load capacity of the bolted joints is not significantly affected by adverse corrosion development in the crevice between the leg member and the splices (see Section 3.1). The results of load tests are very important for steel structure designers and transmission line administrators when evaluating the reliability of in-service steel lattice towers [44]. This finding is confirmed by experience with collapsed transmission towers, where the cause of failure is never the bolted joints of leg members, even though they show crevice corrosion. The collapse is usually caused by critical weakening of the cross-sections of the leg members at the anchorage to the foundations, or by overloading and buckling in strong winds. However, the significant plastic deformation of the splices, caused by the effects of crevice corrosion, is visually unfavourable, and raises concerns among the owners or administrators of the structure about the possible limitation of the bearing capacity or working life of the structure. Rehabilitation of these joints is required for this reason.

It is possible to rehabilitate the connection without dismantling and replacing the splices and bolts with new ones. In order to ensure the longest possible working life of the connection rehabilitated by repair without removing the splices and bolts, it is recommended to use to the following repair procedure [42]:The removal of corrosion products. The crevice between the splice and the web is cleaned of corrosion products to a minimum depth equal to the thickness of the crevice. Cleaning of the external surfaces around the connection is carried out by manual cleaning, or by cleaning with small machinery. In accessible locations, cleaning by blasting, such as high-speed blasting with thermo-blast technology, is recommended to remove corrosion products instead of manual cleaning.The application of the primer, with an overlap of up to 200 mm from the edge of the connection.Sealing the joint. First, it is necessary to seal the inner space of the scratched crevice, then the space between the splice and the leg member is sealed and smoothed, so that the sealant is evenly distributed and does not overlap the splice.Cover the sealant and the overlap from the edge of the connection with a coating (primer and topcoat).

Compatibility between the sealant and the coating system is necessary to achieve the expected life of the rehabilitation system. It is recommended to check the technical condition of the rehabilitation system at regular intervals (optimally, a period of 5 years).

## 5. Conclusions

The paper dealt with the bearing capacity and durability of steel lattice towers made of weathering steels. The main focus was the corrosion damage of the bolted joints of the leg members, and the effect on the load-bearing capacity and durability of the steel structure. Using experimental testing, and on the basis of analytical and numerical calculations, the effect of crevice corrosion on the bolted joint resistance was evaluated. Appropriate design measures, applicable for the rehabilitation of developed crevice corrosion of in-service structures, or the elimination of crevice corrosion in newly designed lattice towers, were evaluated. The main conclusions of the paper are as follows:The capacity of the bolted joint of the leg member is not significantly affected by the development of corrosion products in the crevice, provided that higher strength bolts (8.8 or 10.9) are used, and no significant corrosion weakening of the corner bolt occurs. This finding was verified both by tensile tests on specimens taken from actually operated transmission lattice towers, and by analytical and numerical calculations and static assessments.Using analytical or numerical models, it is possible to determine the static resistance of the joint affected by crevice corrosion with sufficient accuracy. Both calculation procedures given in Section 3.1.1 and Section 3.1.4 can be used in the assessment of structures.The basic design measure is the design of sufficiently rigid splices. It is advisable to design the minimum recommended values for bolt spacing, and the end and edge distances. It is recommended to preload the bolts. This structural recommendation is supported by the results of long-term testing using experimental lattice towers.The application of silicone sealant seems to be the appropriate measure. The measure is effective in the long term, simple to apply, and also relatively cheap. The suitability of this measure is verified for new towers, and also for the rehabilitation of bolted joints in lattice structures in operation.

## Figures and Tables

**Figure 1 materials-15-03397-f001:**
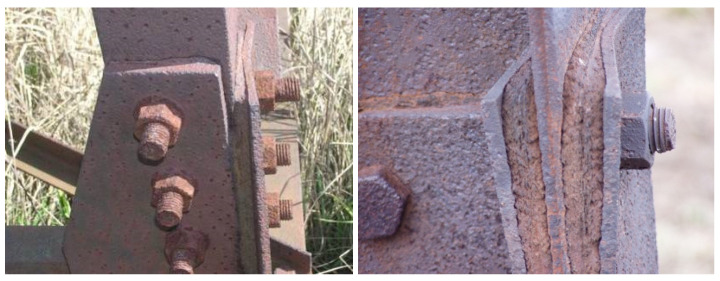
Crevice corrosion in the bolted joint of the leg member.

**Figure 2 materials-15-03397-f002:**
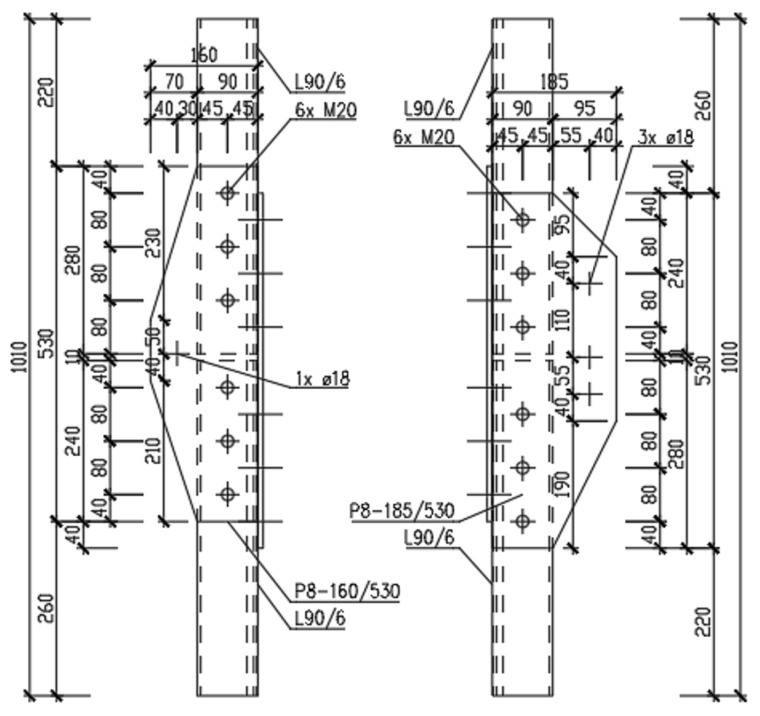
Detail of the A-type joint.

**Figure 3 materials-15-03397-f003:**
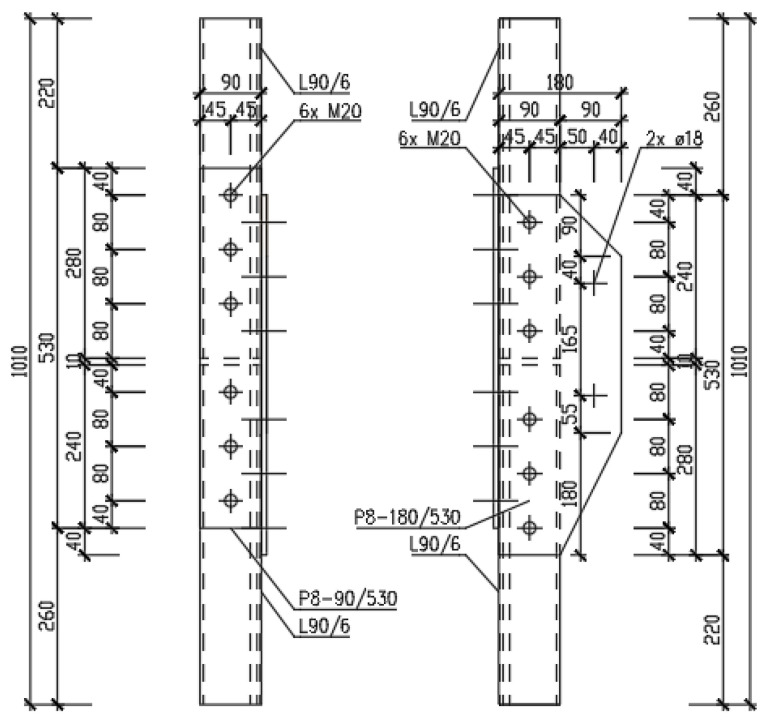
Detail of the K-type joint.

**Figure 4 materials-15-03397-f004:**
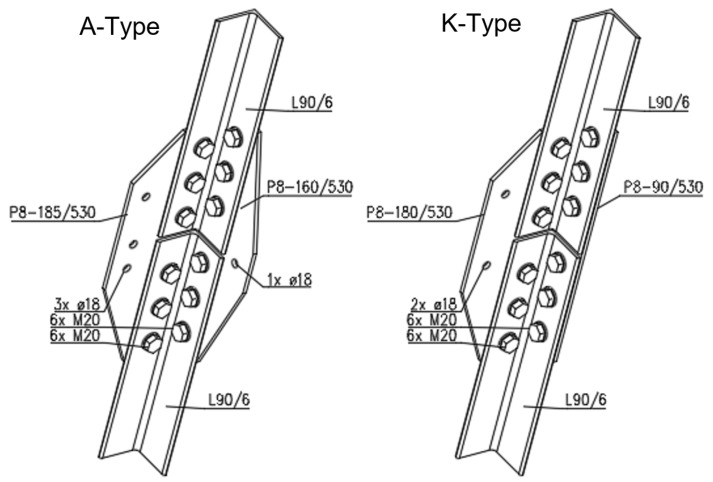
Details of the joint: an isometric view of the joints.

**Figure 5 materials-15-03397-f005:**
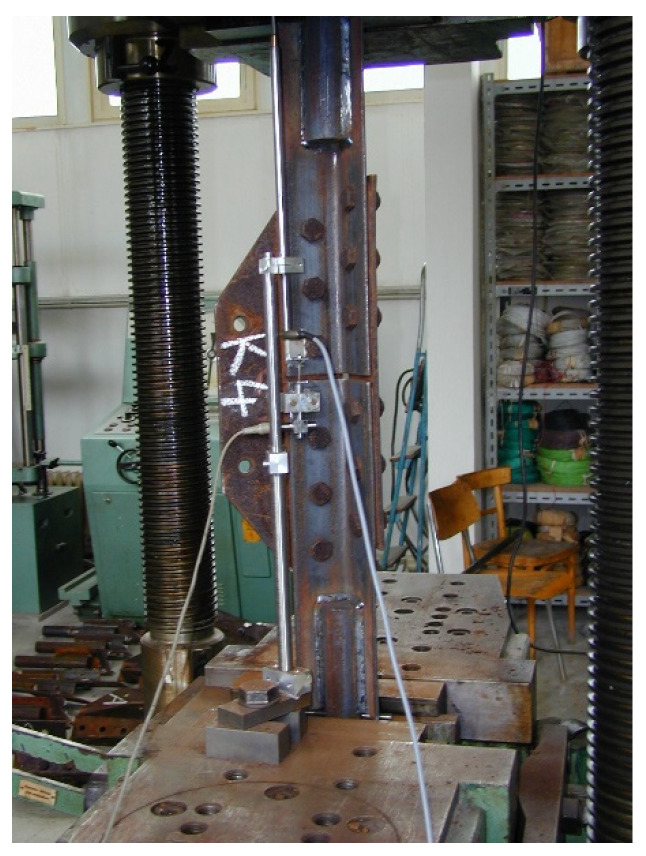
Test piece arrangement in the tensile testing machine.

**Figure 6 materials-15-03397-f006:**
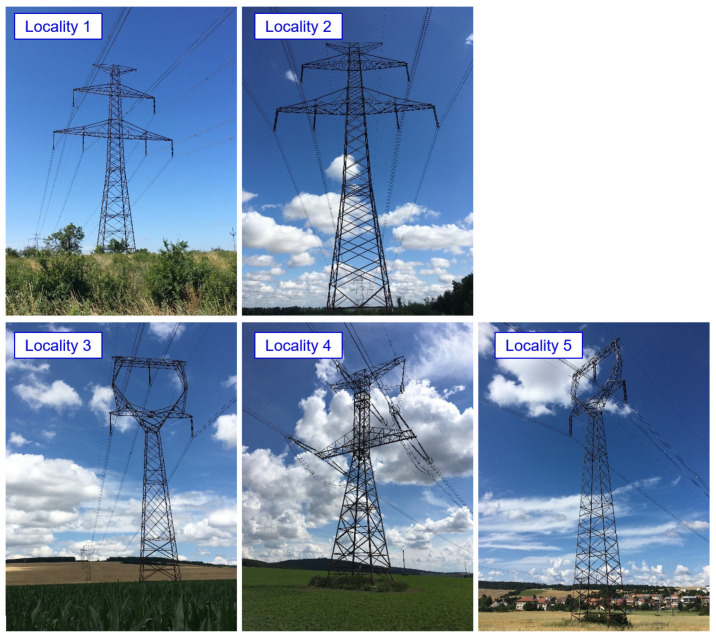
The structural design of the evaluated transmission towers (Locality 1: Tasovice; Locality 2: Dyjákovičky; Locality 3: Ivančice; Locality 4: Rosice; Locality 5: Veverské Knínice).

**Figure 7 materials-15-03397-f007:**
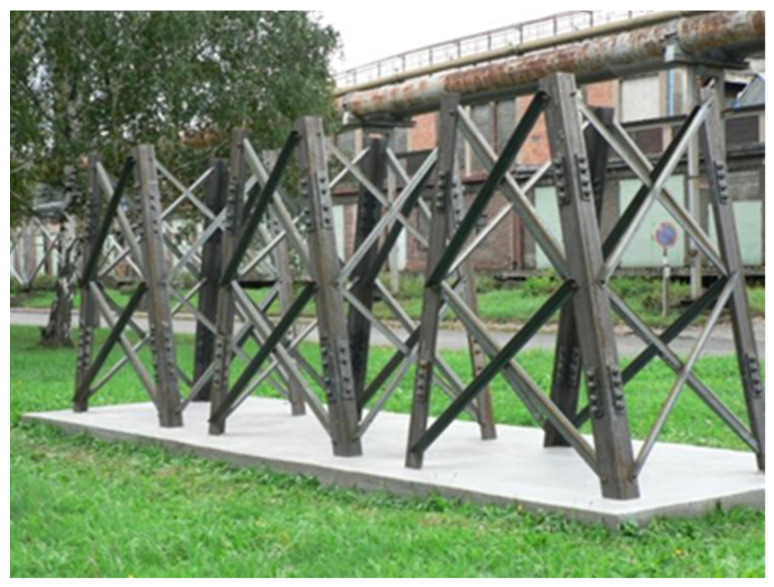
View of the experimental towers’ assembly at the time of installation.

**Figure 8 materials-15-03397-f008:**
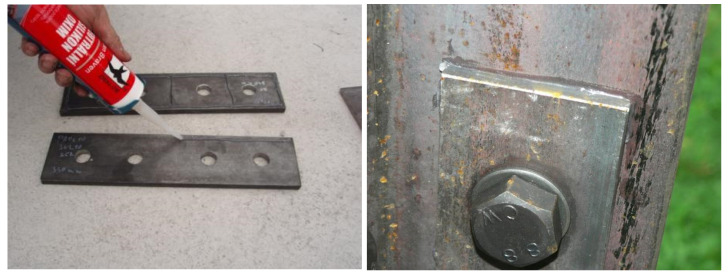
Application of silicone sealant ((**left**) application of sealant to the splices; (**right**) sealant smoothed around the perimeter of the splice).

**Figure 9 materials-15-03397-f009:**
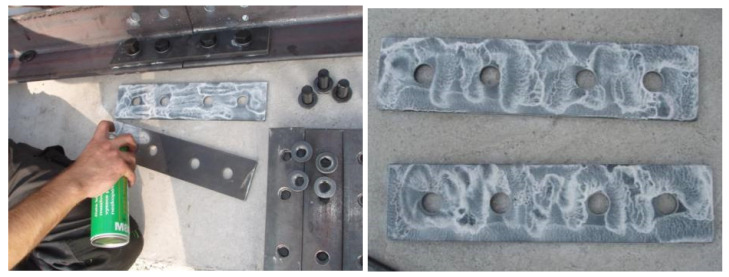
Application of Teflon Vaseline to the splices.

**Figure 10 materials-15-03397-f010:**
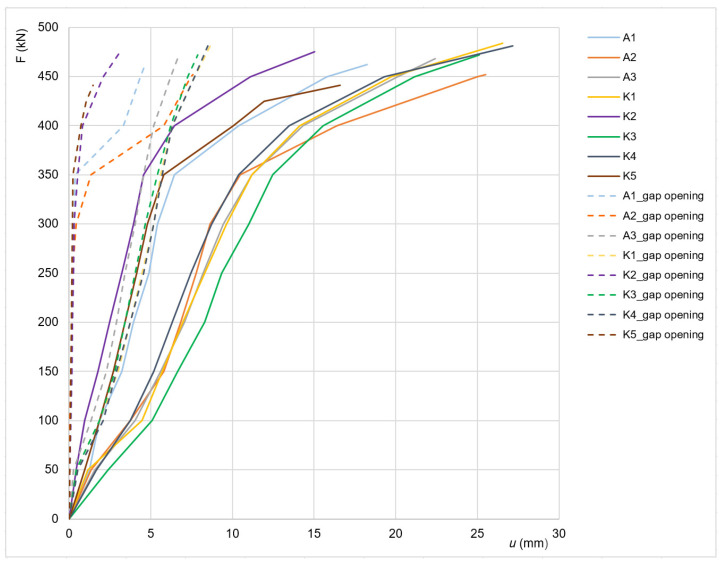
Tensile test results: the solid line shows the dependence of the tensile force *F* (kN) and the total deformation *u* (mm); the dashed line shows the dependence between the tensile force *F* (kN) and the gap opening between the connected angles.

**Figure 11 materials-15-03397-f011:**
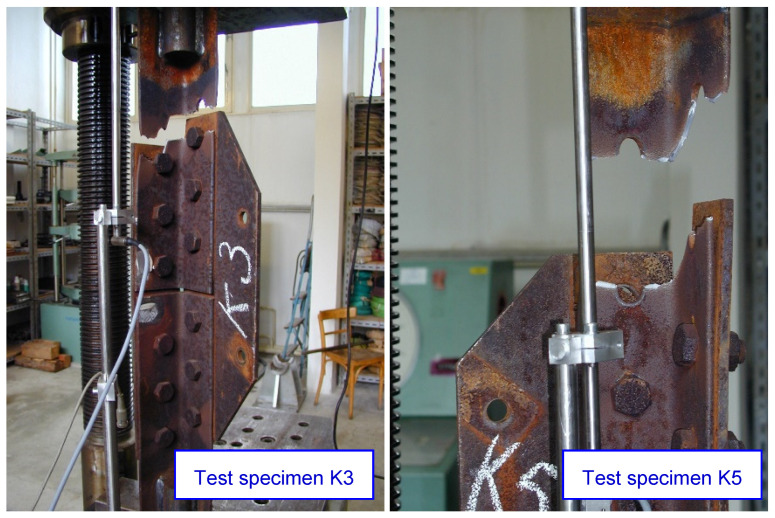
Failure of the test specimens by rupture of the angles (test specimens K3 and K5).

**Figure 12 materials-15-03397-f012:**
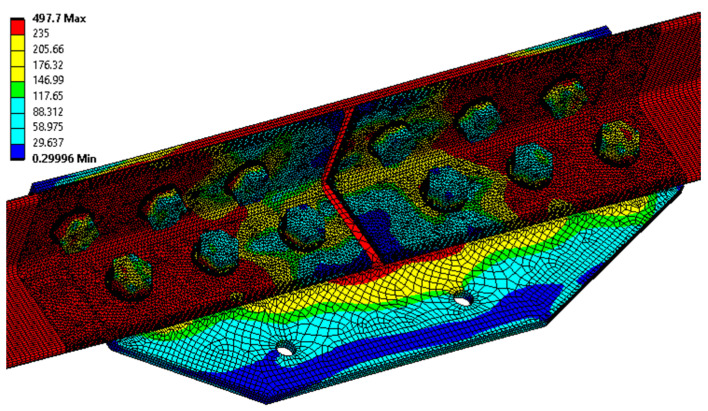
The von Mises equivalent stress (MPa) in the K-type joint (model without the effect of the crevice corrosion).

**Figure 13 materials-15-03397-f013:**
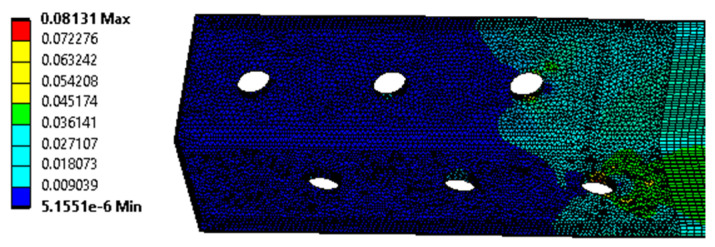
The von Mises equivalent strain in the selected part of the K-type joint (model without the effect of the crevice corrosion).

**Figure 14 materials-15-03397-f014:**
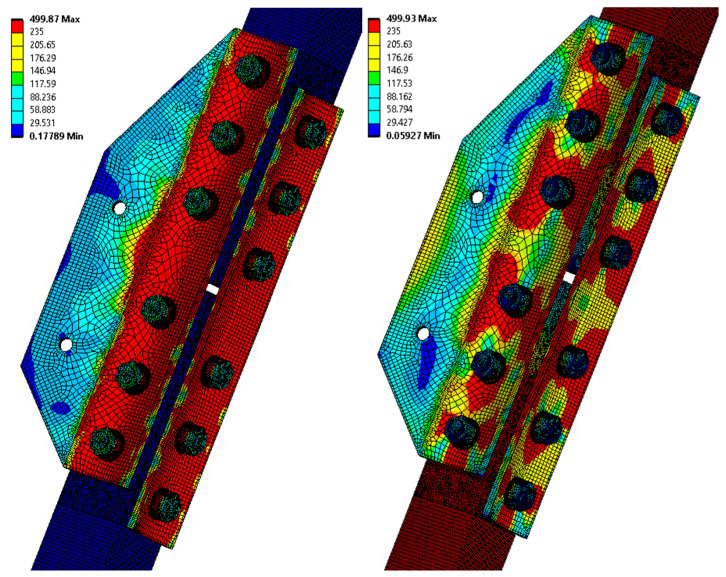
The von Mises equivalent stress (MPa) in the K2 joint (model considering the effect of the crevice corrosion); left: results corresponding to the application of deformations from crevice corrosion; right: results after subsequent loading with the prescribed displacement.

**Figure 15 materials-15-03397-f015:**
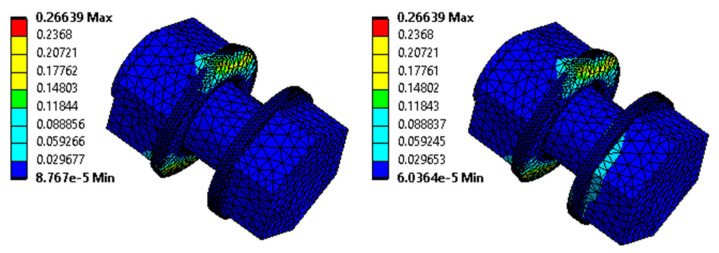
The von Mises equivalent strain in the K2 joint bolts (model considering the effect of the crevice corrosion); left: results corresponding to the application of deformations from crevice corrosion; right: results after subsequent loading with the prescribed displacement.

**Figure 16 materials-15-03397-f016:**
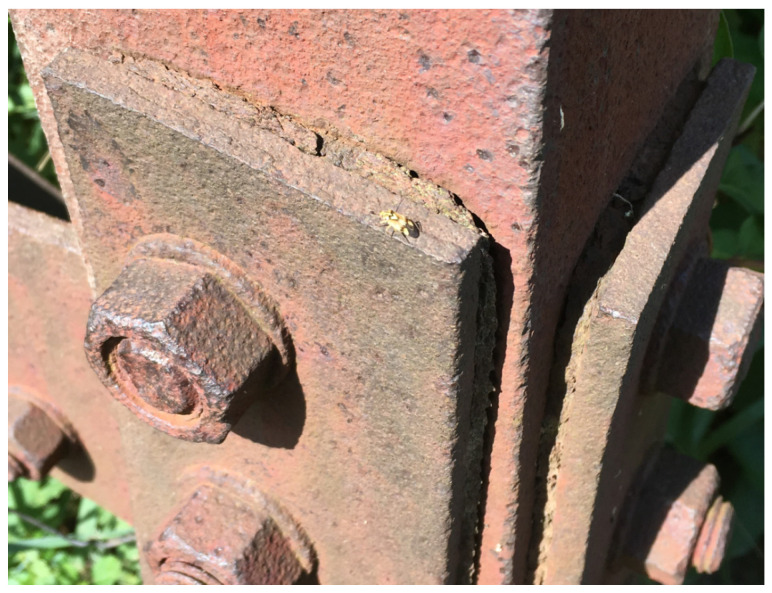
Crevice corrosion at the bolted joint of leg members (Locality 5).

**Figure 17 materials-15-03397-f017:**
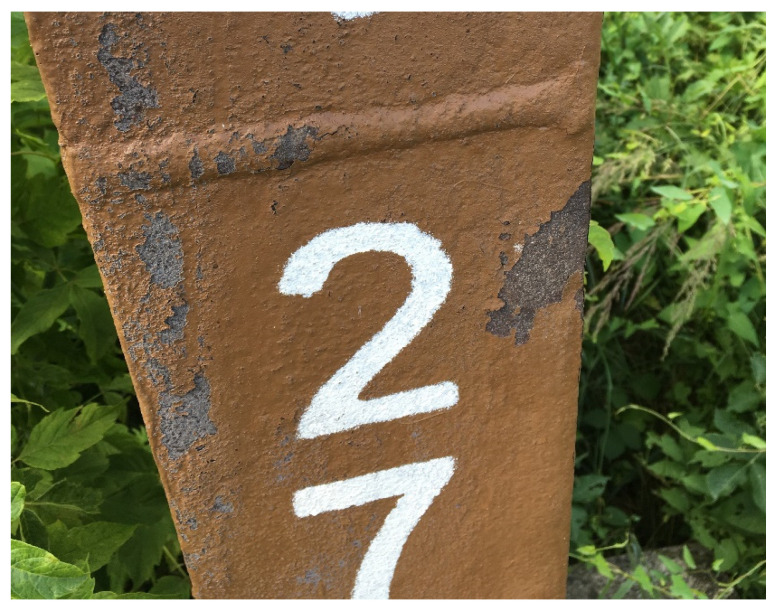
Failure of a paint system applied to a poorly prepared surface (the original protective layer of corrosion products remains under the peeling paint).

**Figure 18 materials-15-03397-f018:**
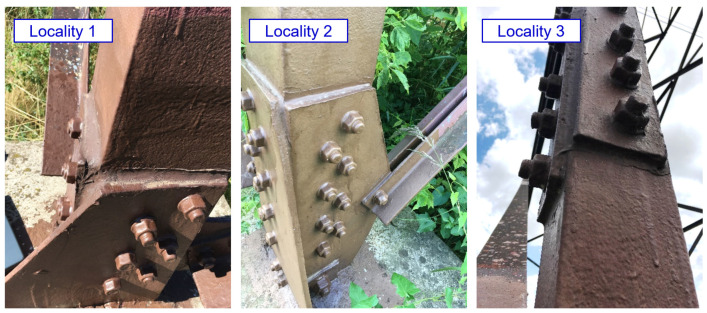
Functional crevice protection with sealant (selected examples).

**Figure 19 materials-15-03397-f019:**
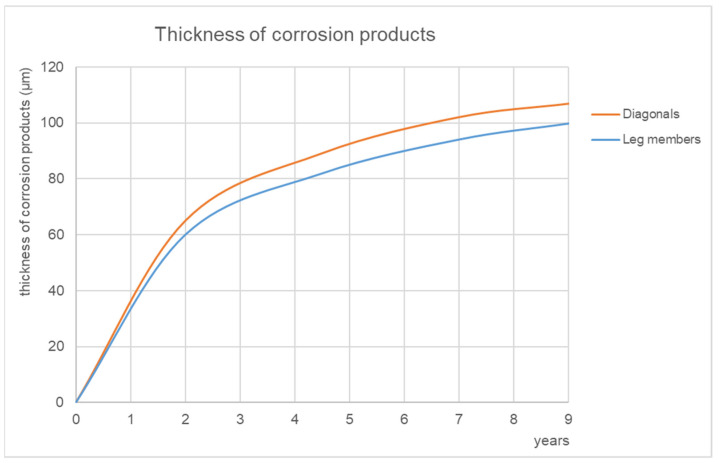
Experimental towers: development of thickness of corrosion products over time.

**Figure 20 materials-15-03397-f020:**
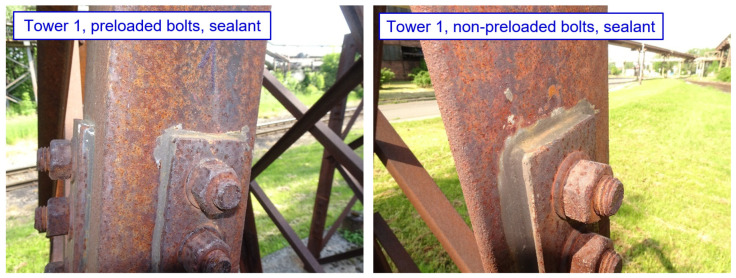
Tower 1: Selected examples of joints (functional sealant along the edge of the splice).

**Figure 21 materials-15-03397-f021:**
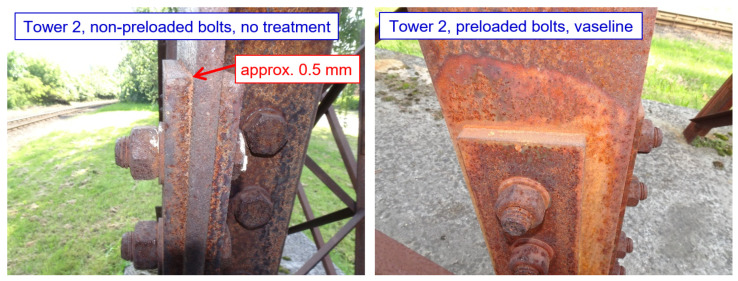
Tower 2: Selected examples of joints ((**left**): initial development of corrosion products in the crevice; (**right**): visual influence of the development of corrosion products in the area around the joints protected with Vaseline).

**Figure 22 materials-15-03397-f022:**
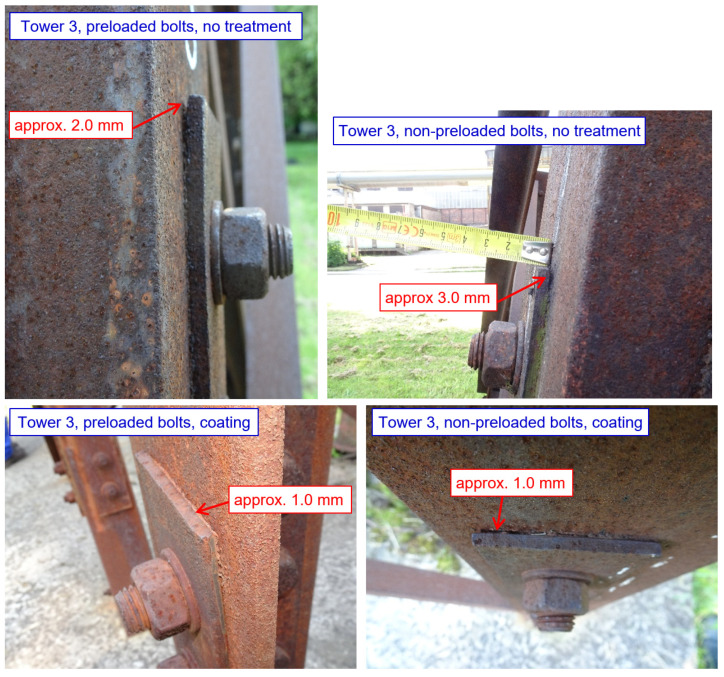
Tower 3: Selected examples of joints (the development of corrosion products in the crevice of joints without treatment, and in joints with coating on the inner surface of the splices).

**Figure 23 materials-15-03397-f023:**
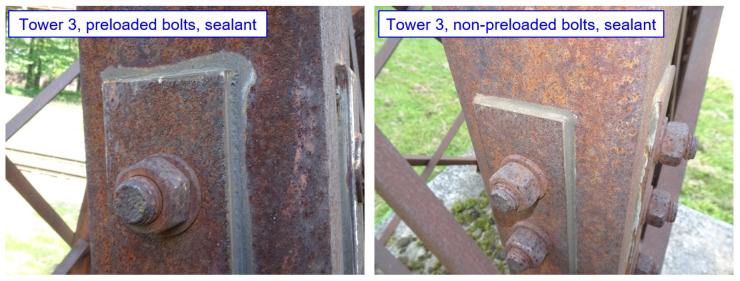
Tower 3: Selected examples of joints (functional sealant along the edge of the splice).

**Figure 24 materials-15-03397-f024:**
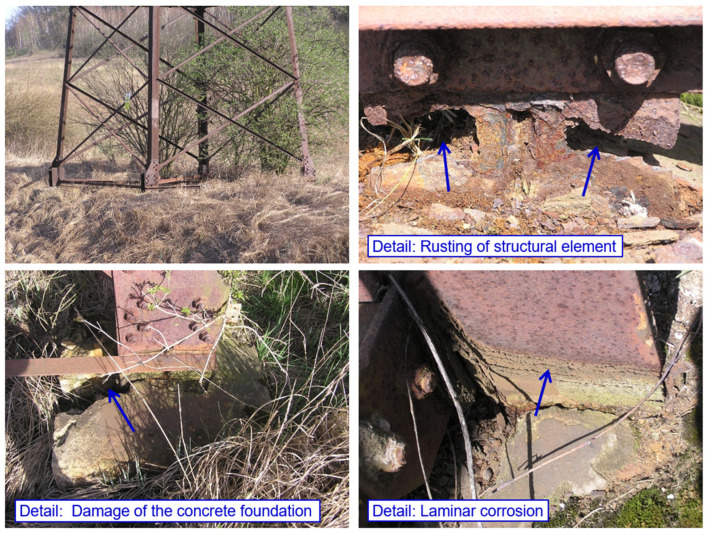
Selected examples of corrosion damage in the anchorage area of transmission towers.

**Table 1 materials-15-03397-t001:** Loading tests of bolted joints: chemical composition of the test pieces.

Standard Test Piece	C	Mn	Si	P	S	Cu	Ni	Cr
EN 10025-5	max 0.19	0.45 1.60	max 0.55	max 0.35	max 0.030	0.20 0.60	max 0.70	0.35 0.85
CSN 41 5217	max 0.14	0.25 1.10	0.22 0.85	0.06 0.18	max 0.040	0.25 0.60	0.25 0.65	0.45 1.35
Leg member No. 1 (L90x6)	0.103	0.48	0.481	0.103	0.012	0.397	0.486	0.849
Leg member No. 2 (L90x6)	0.107	0.56	0.499	0.100	0.018	0.481	0.408	0.702
Splice No. 1 (P8)	0.111	0.59	0.378	0.091	0.009	0.367	0.378	0.810

**Table 2 materials-15-03397-t002:** Loading tests of bolted joints: mechanical properties of test pieces.

Standard Test Piece	Yield Strength *R*_eH_ (MPa)	Tensile Strength *R*_m_ (MPa)	Extension *A*_5_ (%)
EN 10025-5	355	470–630	22
CSN 41 5217	355	490–630	22
Leg member No. 1 (L90x6)	420	504	29.2
Leg member No. 2 (L90x6)	402	529	29.9
Leg member No. 3 (L90x6)	379	500	28.8
Leg member No. 4 (L90x6)	375	524	24.3
Leg member No. 5 (L90x6)	376	517	25.0
Splice No. 1 (P8)	355	505	26.5
Splice No. 2 (P8)	385	515	34.0
Splice No. 3 (P8)	359	497	26.9
minimum measured value	355	497	24.3
average of measured values	381.4	511.4	28.1
maximum measured value	420	529	34.0

**Table 3 materials-15-03397-t003:** Composition of the surface corrosion layer in the crevice (wt. %).

Fe	Cu	Ni	Mn	Cr	P	S	C
36.53	0.04	0.08	0.47	0.05	0.42	1.61	16.40
O	Na	Mg	Al	Si	Cl	K	Ca
38.60	0.83	0.80	1.20	2.13	0.26	0.39	0.19

**Table 4 materials-15-03397-t004:** Composition of the surface corrosion layer in the crevice (atom. %).

Fe	Cu	Ni	Mn	Cr	P	S	C
13.86	0.01	0.03	0.18	0.02	0.29	1.06	28.94
O	Na	Mg	Al	Si	Cl	K	Ca
51.12	0.77	0.70	0.94	1.61	0.16	0.21	0.10

**Table 5 materials-15-03397-t005:** Chemical composition of the leg members.

C	Mn	Si	P	S	Cu	Ni	Cr	Al	V	N
0.11	1.00	0.21	0.11	0.007	0.33	0.17	0.49	0.030	0.04	0.011

**Table 6 materials-15-03397-t006:** Leg members: the mechanical properties of hot-rolled products.

Yield Strength *R*_eH_	Tensile Strength *R*_m_	Extension *A*_5_	Impact Strength KV 7.5 mm (J)		
(MPa)	(MPa)	(%)	*T* = 0 °C	*T* = −20 °C	*T* = −50 °C
427	541	29.7	189	157	71

**Table 7 materials-15-03397-t007:** Diagonals: the mechanical properties of hot-rolled products.

Yield Strength *R*_eH_	Tensile Strength *R*_m_	Extension *A*_5_	Impact Strength KV 7.5 mm (J)		
(MPa)	(MPa)	(%)	*T* = 0 °C	*T* = −20 °C	*T* = −50 °C
419	553	32.9	79	50	-

**Table 8 materials-15-03397-t008:** Splices: the mechanical properties of hot-rolled products.

Member	Yield Strength *R*_eH_	Tensile Strength *R*_m_	Extension *A*_5_	Impact Strength KV 7.5 mm (J)		
	(MPa)	(MPa)	(%)	*T* = 0 °C	*T* = −20 °C	*T* = −50 °C
P80 × 10	409	544	29.7	194	180	-
P60 × 10	395	574	30.3	184	166	-
L80 × 5	438	558	28.9	83	65	-

**Table 9 materials-15-03397-t009:** Summary results of tensile tests.

Test Specimen	*F*_max_ ^1^ (kN)	*u*_max_ ^2^ (mm)	*F*_el_ ^3^ (kN)	*u*_el_ ^4^ (mm)	Failure Mode
A1	462	18.3	~350	6.4	rupture of the angle
A2	452	25.5		10.5	
A3	468	22.4		11.2	
K1	484	26.6		11.2	
K2	475	15.0		4.6	failure of the bolts
K3	472	25.1		12.4	rupture of the angle
K4	481	27.2		10.4	
K5	441	16.6		5.8	
minimum	441.0	15.0	N/A	4.6	N/A
average	466.9	22.1		9.1	
maximum	484.0	27.2		12.4	

^1^ *F*_ma*x*_ … the ultimate load at failure of the test specimen. ^2^ *u*_max_ … total deformation of the test specimen under load *F*_max_. ^3^ *F*_el_ … the elastic capacity of the test specimen. ^4^ *u*_el_ … total deformation of the test specimen under load *F_el_*.

**Table 10 materials-15-03397-t010:** Measured thicknesses of corrosion products.

Locality	Member	*n*	Mean (μm)	Std (μm)	Min (μm)	Max (μm)
Locality 1 GPS: 48.8226420N, 16.1303896E	leg member	30	161.0	37.8	110	226
	leg member	30	151.1	30.5	88	196
	diagonal	30	138.3	39.8	72	254
	diagonal	30	139.5	29.0	74	194
Locality 2 GPS: 48.7637931N, 16.0989825E	leg member	30	197.7	50.8	102	316
	leg member	30	157.1	42.5	90	270
Locality 3 GPS: 49.1171492N, 16.3673119E	leg member	30	167.3	47.2	94	298
	leg member	30	157.1	26.4	100	190
	diagonal	30	178.2	42.0	108	286
	diagonal	30	136.3	32.0	80	192
	diagonal	30	194.0	42.7	130	292
Locality 4 GPS: 49.1847531N, 16.4080061E	leg member	30	165.1	32.1	114	226
	leg member	30	194.0	40.8	116	282
	diagonal	30	150.6	29.9	104	218
	diagonal	30	146.9	27.5	102	192
	diagonal	30	148.9	26.7	100	206
Locality 5 GPS: 49.2328666N, 16.4083097E	leg member	30	170.5	34.7	110	244
	leg member	30	161.8	35.4	112	254
	diagonal	30	154.7	27.1	118	230
	diagonal	30	149.9	35.2	102	238

## Data Availability

Data are contained within the article.

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
