# Peer review of "Corrosion Damage to Joints of Lattice Towers Designed from Weathering Steels"

_materials, 2022, doi:10.3390/ma15093397_

Round 1

Reviewer 1 Report

The manuscript presents the results of a long-term, interesting study concerning the effects of corrosion on bolted joints; the practical and economic implications are relevant. Many results are presented together with useful design guidelines.

Overall considered some aspects of the description of the experimental test and of the finite element analyses need to be improved. The following points should be considered for the revised version of the manuscript:

  • Line 169, please describe the scope of the chemical analysis in reference to your study. What are the implications of this analysis for the work?
  • Line 180-181, the statement “At the same time, the gap opening between the connected angles was measured” is not clear; better explain this concept; a figure may help to support the description.
  • Line 369, it is mentioned Table 7, but the results are reported in Table 9. Table 7 and Table 8 are missing, carefully check the numbering of tables and figures.
  • Line 395, please explain how you modeled the influence of the crevice corrosion.
  • Section 3.1.4, the following aspects regarding the FE model description need to be considered:
    • Did the authors perform a mesh convergence analysis? Give details about the mesh characteristics;
    • What kind of elements were employed? Linear or quadratic elements?
    • How was the contact behavior between bodies simulated? Which algorithm did they use?
    • Figure 14 and Figure 16, only two tetrahedral elements are present along the thickness direction of the plates and the washers. In this way, a poor simulation of the plate bending is obtained and also of the stress state. A proper simulation requires at least three linear brick elements along the thickness direction.
    • Figure 16, specify the direction of the principal strain shown in the contour, or is it a Von Mises equivalent strain?
  • The introduction section should also consider some works related to the FEA of bolted joints such as: https://doi.org/10.1016/j.engstruct.2021.113368; https://doi.org/10.1016/j.compstruct.2020.113199

Author Response

Authors of the article would like to thank the reviewer for valuable comments and recommendations. The authors have tried to include all recommendations in the revised version of the article.

The manuscript presents the results of a long-term, interesting study concerning the effects of corrosion on bolted joints; the practical and economic implications are relevant. Many results are presented together with useful design guidelines.

Overall considered some aspects of the description of the experimental test and of the finite element analyses need to be improved. The following points should be considered for the revised version of the manuscript:

  • Line 169, please describe the scope of the chemical analysis in reference to your study. What are the implications of this analysis for the work?

Authors of the article thank the reviewer for a useful comment. The following text has been added to the article:

Chemical analysis has been carried out for two selected angles (test specimen A1 and K1) and one splice (test specimen K1). The chemical analysis confirmed that the specimens taken from the collapsed lattice towers correspond to standard weathering steels in accordance with the requirements of the standards [34, 35].

  • Line 180-181, the statement “At the same time, the gap opening between the connected angles was measured” is not clear; better explain this concept; a figure may help to support the description.

The authors thank the reviewer for a relevant comment. Because the second reviewer recommended limiting the number of figures without significant scientific significance, the authors decided to provide a verbal explanation:

gap opening (i.e. change of the longitudinal distance between the ends of the connected angles)

  • Line 369, it is mentioned Table 7, but the results are reported in Table 9. Table 7 and Table 8 are missing, carefully check the numbering of tables and figures.

The authors thank you for this notice. Tables in the article have been renumbered.

  • Line 395, please explain how you modelled the influence of the crevice corrosion.

The following explanation is given in the article:

The consideration of the effect of crevice corrosion was based on a specific case of deformation of the splice of the K2 tested joint, where the measured maximum size of the crevice with corrosion products was equal to 10 mm. The model prescribes a 10 mm deformation of the vertical lines of the splices away from the L profile.

  • Section 3.1.4, the following aspects regarding the FE model description need to be considered:
    • Did the authors perform a mesh convergence analysis? Give details about the mesh characteristics;

The authors thank the reviewer for this comment. Based on the recommendations, a convergence analysis was performed with a gradual refinement of the mesh. A mesh with an average element size of 2 mm was found to be optimal. Further refinement of the mesh did not lead to statically significant changes in the observed static quantities.

    • What kind of elements were employed? Linear or quadratic elements?

Linear finite elements were used in the numerical model.

  • How was the contact behavior between bodies simulated? Which algorithm did they use?

For contact between bodies was used the ANSYS contact tool with friction (coefficient of friction 0.15).

  • Figure 14 and Figure 16, only two tetrahedral elements are present along the thickness direction of the plates and the washers. In this way, a poor simulation of the plate bending is obtained and also of the stress state. A proper simulation requires at least three linear brick elements along the thickness direction.

The calculation and the figures have been adjusted for the refined mesh.

    • Figure 16, specify the direction of the principal strain shown in the contour, or is it a Von Mises equivalent strain?

Both the description and the figure have been modified (von Mises equivalent strain)

  • The introduction section should also consider some works related to the FEA of bolted joints such as: https://doi.org/10.1016/j.engstruct.2021.113368; https://doi.org/10.1016/j.compstruct.2020.113199

The authors agree with the reviewer's recommendation. The Introduction has been expanded to include the following text (relevant references have also been added):

Calculation analyses of bolted joints can be performed using analytical relations according to current standards [24, 25]. For detailed numerical analyses of bolted joints using the finite element method, the recommendations given in technical publications [26, 27, 28] can be used.

Reviewer 2 Report

The article is about corrosion damage to joints of lattice towers designed from weathering steels. However, some issues must to be addressed:

  1. Abstract: Please start by expressing the aim of this paper, followed by the rest of the information. Typically, the abstract should provide a broad overview of the entire project, summarize the results, and present the implications of the research or what it adds to its field.
  2. The introduction and references sections must to severe improved since the scientific literature in related files is missing!
  3. Instead of “leg member” the authors must to find a technical definition in order to comply with high level of the journal.
  4. Figure 1, 2, 7, 9, 10, 17, 18, 21, 24, 25, 26-29 has no scientific meaning: the aspects presented can be better described in a paragraph each place where is needed. The pictures are presenting some technical issues without a scientific novelty in order to increase the quality of the article.
  5. The calculus from 3.1.1. and 3.1.2. must to be shorten in order to provide only the most important aspects.
  6. Figure 12: please indicate details in figure for a better readability of the article.
  7. Somewhere in figures the authors mention locality 1 to 5 and in some tables they uswe the name of the localities …
  8. The results are merely presented, not properly discussed. Please add explanations for the observed changes. Please give an extended discussion on the obtained results and correlate your findings with previous literature studies and prospective applications.
  9. More analysis and interpretation of the results should be added for a clearer understanding of observed experimental phenomena.
  10. The authors must to provide some details about importance of the research and their applicability.
  11. Please enhance the clarity of the conclusion section in order to highlight the results obtained.
  12. General check-up and correction of the English language is suggested. There are still some minor typos and grammatical errors.

The author needs to address the abovementioned points for the betterment of the manuscript.

Author Response

Authors of the article would like to thank the reviewer for valuable comments and recommendations. The authors have made an effort to incorporate the recommendations into the text, while respecting the main aims and structure of the article (see comment on point 11). Detailed responses are provided below for each of the reviewer's recommendations.

The article is about corrosion damage to joints of lattice towers designed from weathering steels. However, some issues must to be addressed:

  1. Abstract: Please start by expressing the aim of this paper, followed by the rest of the information. Typically, the abstract should provide a broad overview of the entire project, summarize the results, and present the implications of the research or what it adds to its field.

Authors of the article thank the reviewer for a relevant comment. The abstract has been edited.

  1. The introduction and references sections must to severe improved since the scientific literature in related files is missing!

Authors of the article thank the reviewer for this recommendation. The Introduction has been modified to reflect the scientific literature.

  1. Instead of “leg member” the authors must to find a technical definition in order to comply with high level of the journal.

The terminology "leg member" is taken from the current standards, specifically from "EN 1993-3-1: Eurocode 3 - Design of steel structures - Part 3-1: Towers, masts and chimneys - Towers and masts". The members of lattice towers are divided into two groups “leg members” and “bracing members”. The authors thank the reviewer for pointing this out and realize that this classification may not be universally known. Therefore, for the first use of the term "leg member", a reference has been inserted referring to EN 1993-1-3.

  1. Figure 1, 2, 7, 9, 10, 17, 18, 21, 24, 25, 26-29 has no scientific meaning: the aspects presented can be better described in a paragraph each place where is needed. The pictures are presenting some technical issues without a scientific novelty in order to increase the quality of the article.

Authors of the article thank the reviewer for this comment. We generally agree with this comment. However, the authors have tried to prepare an article that will be useful not only for the scientific community, but also for construction practitioners. By means of photographs, the authors try to explain the issues better. Based on the reviewer's recommendation, a re-evaluation has been made and five figures (Fig. 1 and Figs. 26, 27, 28, 29) have been removed from the article, leaving only the reference to the literature. The authors decided to keep the other figures for clarity of the article

  1. The calculus from 3.1.1. and 3.1.2. must to be shorten in order to provide only the most important aspects.

The authors have respected the reviewer's comment. There has been a significant reduction in calculations.

  1. Figure 12: please indicate details in figure for a better readability of the article.

The authors thank the reviewer for a relevant comment. The figure has been modified.

  1. Somewhere in figures the authors mention locality 1 to 5 and in some tables they use the name of the localities …

The authors thank the reviewer for his comments. The naming of the localities has been unified.

  1. The results are merely presented, not properly discussed. Please add explanations for the observed changes. Please give an extended discussion on the obtained results and correlate your findings with previous literature studies and prospective applications.

The authors thank for this recommendation. Some of the phenomena discussed in Chapter 4 are explained in more detail after modification, also with respect to the available previous literature studies.

  1. More analysis and interpretation of the results should be added for a clearer understanding of observed experimental phenomena.

The authors thank for this recommendation. The following is an explanation and suggested changes to the article.

For the experimental lattice towers, only a visual evaluation of corrosion product development, measurement of corrosion product thickness on the surface of the elements, measurement of crevice opening and measurement of real thicknesses of the steel elements were carried out during the 9 years of exposure. This information has been added to the relevant parts of the paper (section 2.2, section 3.3.1). As the experimental lattice towers project is designed as a long-term project (assumed to be at least 25 years), detailed analyses of the corrosion products taken from the crevice of the dismantled bolted joints are not yet available.

The results of the analysis of the composition of the corrosion layer from the crevice surfaces of the tested joints taken from the crashed transmission towers have been added to the article and discussed (results added to section 2.1). Detailed information on the composition of the patina on the samples from the crashed transmission towers is also available. However, this information is not presented in the article as the focus is mainly on the effect of corrosion damage on the load carrying capacity of the joints.

  1. The authors must to provide some details about importance of the research and their applicability.

The authors of the article thank for this relevant comment. The explanation "importance of the research and their applicability" has therefore been inserted at the end of section 1.The main target readership of the article is mainly experts from construction practice who have to deal with the issue of the influence of corrosion on the load-bearing capacity and durability of steel structures. The structure of the whole article is subordinated to this goal, where detailed information about the microstructure of corrosion products is deliberately not given, as this information is not relevant for the target group of readers. More important are the results of static tests, analytical and numerical calculations and also the stage of practical effects of the tested structural modifications of the bolted joints.

  1. Please enhance the clarity of the conclusion section in order to highlight the results obtained.

The authors thank for this recommendation. Conclusion section has been edited.

  1. General check-up and correction of the English language is suggested. There are still some minor typos and grammatical errors.

The main issue is explained in point 3. The article will be sent for external English proofreading.

The author needs to address the abovementioned points for the betterment of the manuscript.

Round 2

Reviewer 1 Report

All the review issues were improved by the authors.

In my opinion, the paper can now be accepted for publication in Materials.

Reviewer 2 Report

The article was improved and can be published.